Hibiscetin attenuates lipopolysaccharide-evoked memory impairment by inhibiting BDNF/caspase-3/NF-κB pathway in rodents

Gilani Sadaf Jamal 1
http://orcid.org/0000-0002-3017-9924 Bin Jumah May Nasser 2 3 4
Fatima Farhat 5
Al-Abbasi Fahad A. 6
Afzal Muhammad 7
http://orcid.org/0000-0003-4007-4023 Alzarea Sami I. 8
Sayyed Nadeem 9
Nadeem Muhammad Shahid 6 mhalim@kau.edu.sa
http://orcid.org/0000-0003-1881-5219 Kazmi Imran 6 ikazmi@kau.edu.sa
1 Department of Basic Health Sciences, Foundation Year, Princess Nourah bint Abdulrahman University , Riyadh , Saudi Arabia
2 Environment and Biomaterial Unit, Health Sciences Research Center, Princess Nourah bint Abdulrahman University , Riyadh , Saudi Arabia
3 Saudi Society for Applied Science, Princess Nourah bint Abdulrahman University , Riyadh , Saudi Arabia
4 Biology Department, College of Science, Princess Nourah bint Abdulrahman University , Riyadh, Riyadh , Saudi Arabia
5 Department of Pharmaceutics, College of Pharmacy, Prince Sattam Bin Abdulaziz University , Al-kharj , Saudi Arabia
6 Department of Biochemistry, Faculty of Sciences, King Abdulaziz University , Jeddah , Saudi Arabia
7 Department of Pharmaceutical Sciences, Pharmacy Program, Batterjee Medical College , Jeddah , Saudi Arabia
8 Department of Pharmacology, College of Pharmacy, Jouf University , Sakaka , Saudi Arabia
9 School of Pharmacy, Glocal University , Saharanpur, Uttar Pradesh , India
Khosravi Mohsen
Electronic publication date: 2024 Jan 31
Publication date: 2024
Volume: 12
Electronic Location ID: e16795
Received 2023 May 15; Accepted 2023 Dec 24
Copyright: © 2024 Gilani et al.
Copyright year: 2024
Copyright holder: Gilani et al.
License: This is an open access article distributed under the terms of the Creative Commons Attribution License, which permits unrestricted use, distribution, reproduction and adaptation in any medium and for any purpose provided that it is properly attributed. For attribution, the original author(s), title, publication source (PeerJ) and either DOI or URL of the article must be cited.
License URL: https://creativecommons.org/licenses/by/4.0/

Keywords: Hibiscetin, Lipopolysaccharide, Memory, Neuroinflammatory, Oxidative stress

Funding: Princess Nourah bint Abdulrahman University Researchers PNURSP2023R108 This research project was supported by the Princess Nourah bint Abdulrahman University Researchers Supporting Project number (PNURSP2023R108), Princess Nourah bint Abdulrahman University, Riyadh, Saudi Arabia. The funders had no role in study design, data collection and analysis, decision to publish, or preparation of the manuscript.

==============================
This study explores the neuroprotective potential of hibiscetin concerning memory deficits induced by lipopolysaccharide (LPS) injection in rats. The aim of this study is to evaluate the effect of hibiscetin against LPS-injected memory deficits in rats. The behavioral paradigms were conducted to access LPS-induced memory deficits. Various biochemical parameters such as acetyl-cholinesterase activity, choline-acetyltransferase, antioxidant (superoxide dismutase, glutathione transferase, catalase), oxidative stress (malonaldehyde), and nitric oxide levels were examined. Furthermore, neuroinflammatory parameters such as tumor necrosis factor-α, interleukin-1β (IL-1β), IL-6, and nuclear factor-kappa B expression and brain-derived neurotrophic factor as well as apoptosis marker i.e., caspase-3 were evaluated. The results demonstrated that the hibiscetin-treated group exhibited significant recovery in LPS-induced memory deficits in rats by using behavioral paradigms, biochemical parameters, antioxidant levels, oxidative stress, neuroinflammatory markers, and apoptosis markers. Recent research suggested that hibiscetin may serve as a promising neuroprotective agent in experimental animals and could offer an alternative in LPS-injected memory deficits in rodent models.

Introduction

Neurodegenerative diseases are a diverse group of conditions that are characterized by the progressive loss of neurons in specific regions of the brain. This neuronal loss leads to a variety of clinical manifestations, including impairments in mobility, coordination, strength, sensation, and cognition. Common neurodegenerative diseases include Alzheimer’s disease (AD), Huntington’s disease (HD), and Parkinson’s disease (PD) (Alam & Krishnamurthy, 2022; Amraie, Pouraboli & Rajaei, 2020; Kou et al., 2022). These diseases can manifest at any age but are most commonly diagnosed in the elderly as they progress. A hallmark feature of AD is memory impairment, which can render patients incapable of performing even mundane tasks. The disease is further exacerbated by oxidative stress and the release of neuroinflammatory mediators (Azmand & Rajaei, 2021; Babaei et al., 2022).

Lipopolysaccharide (LPS), a glycolipid component of the outer membrane of Gram-negative bacteria, triggers neuronal death, inhibits neurogenesis, disrupts synaptic plasticity, and impairs memory (Zhu et al., 2014). Peripheral administration of LPS has been shown to result in functional impairments in the brain. LPS-induced peripheral infection triggers the immune system to produce inflammatory cytokines in the brain (Zakaria et al., 2017). The neuroinflammatory response triggered by LPS releases various parameters that exacerbate the disease (Zhou et al., 2021; Zolfaghari, Rabbani Khorasgani & Noorbakhshnia, 2021; Careena et al., 2018; da Rosa et al., 2022; Fachel et al., 2020; Fang et al., 2019). There are millions of people worldwide who suffer from neurodegenerative diseases. Natural alternatives to synthetic drugs are gaining popularity as treatments for most of these extremely complex neurological conditions (Sadraie et al., 2019; Schilling et al., 2021).

Hibiscetin obtained from Hibiscus safdariffa, Sadum album, and Lotus japonecus is a flavonoid and has been supposed to improve memory impairment and balance the various associated parameters in HD and similar conditions (Norouzi et al., 2019; Pierre et al., 2022; Pusceddu et al., 2022). Hibiscetin is a naturally occurring flavonoid with anti-inflammatory and antioxidant properties. It has also been shown to have neuroprotective effects by modulating the expression of genes involved in synaptic plasticity, neurogenesis, and memory formation (El-Shiekh et al., 2020). Previous studies have demonstrated the antioxidant activity of hibiscetin (Ragi & Muraleedharan, 2023), as well as its efficacy against streptozotocin-induced diabetes (Gilani et al., 2022), and rotenone-induced PD and 3-nitropropionic acid-induced HD (Alzarea et al., 2022; Mahdi et al., 2023) by inhibiting inflammatory and oxidative stress. Therefore, hibiscetin may be a promising therapeutic candidate for neurological disorders (El-Shiekh et al., 2020; Afolabi et al., 2023). This is the first study to investigate the potential use of hibiscetin as a therapeutic agent in the context of memory impairment induced by LPS in rodents via behavioral activity such as the Morris water maze and Y-maze test. Additionally, the molecular mechanisms of hibiscetin action were investigated by measuring the levels of inflammatory cytokines, oxidative stress markers, and several molecular pathways, including brain-derived neurotrophic factor (BDNF), nuclear factor-kappa B (NF-κB), and caspase-3 in the rat.

Methodology

Animals

During the experiment, Wistar rats (180 ± 20 g) (male 10–12 weeks old) rats were procured and provided with the facilities of animal care, feeding, housing, and enrichment from the animal center and research laboratory of T.G. Services, M.S., India. The rats were maintained under standard conditions with a 12 × 12-h light-dark cycle, a temperature of 24 ± 2 °C, and humidity control at 50% to 60%. The rats received a diet comprising 60% of the total calories from fat and had access to water ad libitum. In the laboratory, rats were acclimatized for 7 days under these situations and the animals were included in the study, without any prior procedures. An Institutional Animal Care and Use Committee (IACUC/TRS/PT/022/027) in India approved the rodent experiments and research was conducted in accordance with ARRIVE guidelines (Percie du Sert et al., 2020).

Drugs and chemicals

LPS (Escherichia coli 0111: B4, MSW-LPS) and hibiscetin (98.0%, MSW-H01) were purchased from MSW Pharma, M. S., India. The levels of interleukin-6 (IL-6), IL-1β, NF-κB, BDNF, and caspase-3 were measured using ELISA kits (MSW Pharma, M.S., India). The other chemicals involved in the experiments were analytical standards.

Research design

Twenty-four rats were randomly assigned to four groups (n = 6): control (saline), LPS control, LPS + hibiscetin (10 mg/kg p.o.), and hibiscetin per se (10 mg/kg p.o.). LPS (1 mg/kg) was administered intraperitoneally for 7 days, with hibiscetin (10 mg/kg p.o.) administered 1 h after LPS injection. Animals were assessed using the Morris water maze and Y-maze tests. This immediately followed behavioral assessments and euthanizing animals at the end of the experiment. Subsequently, the rats were sacrificed for estimation of acetylcholine esterase (AChE), choline-acetyltransferase (ChAT), oxidative stress (Malondialdehyde-MDA), antioxidant (catalase activity-CAT, superoxide dismutase-SOD, glutathione-GSH), nitric oxide stress, proinflammatory markers (IL-6, IL-1β, TNF-α, NF-κB), BDNF, and caspase-3.

Acute toxicological study

According to OECD guideline no.423, an acute toxicity study was conducted. The rats were monitored for signs of toxicity over the next 14-day. As previously reported, hibiscetin was administered orally at the recommended safe dose (Gilani et al., 2022; Mahdi et al., 2023; Alzarea et al., 2022). Clinical symptoms were examined, such as behavioral, vision, weight, skin, and fur changes.

Behavioral parameters

Y-maze test

The working memory of the animals was assessed using a Y-maze. The random rearrangements that occurred during the test were recorded. The maze was 40 cm long, 35 cm high, and 12 cm wide and consisted of three wooden arms with an offset of 120° on each side. In addition, each tower had a unique design on its walls, collectively referred to as X, Y, and Z. In the maze, rats were allowed to freely explore each arm to find the way out, which was divided into sections. The experiment was conducted over 5 min to count the number of visits to each arm. After each session, the apparatus was cleaned with a 10% ethanol solution to eliminate odors. A random distribution alternates a three-arm combination of three consecutive entries, for example, XYZ, ZXY, and YZX. Spontaneous alternation performance (SAP) was used to calculate SAP scores and SAP percentages (Alshehri & Imam, 2021; Ammari et al., 2018; Kupferschmid, Rowsey & Riviera, 2020).

Morris water maze test

The Morris water maze test involved a round tank and had four even quadrants or zones. Each quadrant was equipped with a hidden platform located 1 cm below the water surface for the first five days. Small white particles were scattered across the water’s surface. In each learning session, a randomly selected rat was placed in a random tank location. The average escape latency was calculated when the rat touched and covered the platform. If the rat took longer than 60 s to reach the platform, the latency was noted. A 20-s interval was allowed between sessions. Each animal was delicately wiped down and put into its corresponding home after passing the tests at all three entry points. Five days after the trial began; rats were tested on their ability to remember the hidden platform’s location for 60 s. Platforms were removed from tanks during this phase. A second measurement was the latency time to locate the proper quadrant as well as the time spent in the compartment (Hassanzadeh-Taheri et al., 2021; He et al., 2021; Hosseini, Salmani & Baghcheghi, 2021).

Estimation of biochemical parameters

Homogenization of brain tissue

Rats were anesthetized with ketamine (75 mg/kg) and xylazine (10 mg/kg) intraperitoneally and sacrificed by cervical dislocation. Then, their brains were isolated and cleaned with ice-cold saline. The brain samples were suspended in a pH 7 buffer solution and centrifuged at 1,000 rpm for 10 min. The supernatant was collected and used for biochemical analyses.

Detection of AChE and ChAT

The mixture consisted of acetylcholine iodide (1 mM), DTNB (2 mM), potassium phosphate buffer (100 mM) of pH 7, and homogenate of the hippocampus/striatum in a total volume of 500 L. This mixture was subjected to incubation (10 min at 37 °C). The reaction was ended by adding serine hemisulphate (0.5 mM). A yellow colour solution was obtained which was analyzed at λmax 412 nm determining the AChE levels and measured as mmole/min/g (Ellman et al., 1961; Jangra et al., 2021). ChAT activity was estimated with the aid of commercially available kits and concentration was measured in U/g.

Prediction of SOD

To measure SOD activity, nitroblue tetrazolium (NBT) was applied through photochemical means. Each reaction mixture, with a volume of 1.5 mL, contained 100 mM Tris/HCl (pH 7.8), 75 mM NBT, 2 M riboflavin, and 6 mM EDTA. To this, 1.5 mL of supernatant was added. A difference in absorbance at 560 nm caused by formazan formation was calculated. and SOD activity was expressed in U/mg (Arab et al., 2022).

Measurement of reduced glutathione

A total of 0.1 M phosphate buffer at pH 7.4 was used to homogenate of brain tissues to determine glutathione (GSH) concentrations. The homogenates were precipitated with a solution of trichloroacetic acid containing 1 mM EDTA at a concentration of 20%. A 5-min stand time was allowed after centrifuging the mixture. The supernatant (200 μl) was then transferred to freshly cleaned test tubes and added 1.8 ml of Ellman’s reagent (5, 5′-dithio bis-2-nitrobenzoic acid 0.1 mM) was prepared in 0.3 M phosphate buffer with 1 % sodium citrate solution). Then all the test tubes were filled to a volume of 2 mL. At the end, the solutions were measured at 412 nm against a blank. The concentration was measured in U/mg (Arab et al., 2022).

Catalase estimation

The reaction mixture contained 100 mL of supernatant and 150 mL of buffer (phosphate), 0.01 M, pH 7.0. 250 L of H2O2 0.16 M was added to the reaction and allowed to run for 1 min at 37 °C before being stopped by the addition of 1.0 mL of a dichromate acetic acid reagent. The tubes were placed in a hot water bath for 15 min. A green color produced at the end was determined at a wavelength of 570 nm. Moreover, parallel processing was done using tubes without any enzyme. The catalase (CAT) activity was determined in U/mg by measuring the change in absorbance per unit (Arab et al., 2022).

Determination of malondialdehyde

The test tubes were filled with 500 ml of homogenate and the control tube with 500 μl Tris-HCl buffer (50 mM, pH 7,4). In each tube, 250 ml 20% trichloroacetic acid (TCA), and 500 μl of 0.67% TBA were added. The tubes were closed with glass beads and incubated in a water bath for 10 min at 90 °C. After cooling at room temperature, they were centrifuged for 15 min at 3,000 rpm. Absorbance was calculated at 530 nm and amounts were estimated as µmol/L (Arab et al., 2022).

Nitric oxide estimation

The oxidation of nitric oxide (NO) results in the formation of nitrite, which is used to calculate nitric oxide. An assay for nitrite formed after oxidizing nitric oxide was conducted as mentioned in diazotization reactions (Amraie, Pouraboli & Rajaei, 2020; Amraie et al., 2022; Andy et al., 2018).

Neuroinflammatory cytokines

The quantification of IL-1β, IL-6, TNF-α, and NF-κB was performed by standard ELISA kits and as per the standard procedures. The amounts of IL-1β, IL-6, and TNF-α were estimated as pg/mL. The concentration of NF-κB was measured in ng/mL.

BDNF and caspase-3 activity

The homogenized brain was treated with antibodies against BDNF to measure the concentration of BDNF. ELISA technique was used for the analysis of caspase-3 activity. The BDNF and caspase-3 were evaluated by using standard assay kits.

Statistical analysis

The values of results were analyzed using GraphPad Prism-9 (GraphPad Software Inc., La Jolla, CA, USA) and represented as mean ± SEM. The normality pass test was executed using the Shapiro-Wilk Test. The data were examined for the MWM test by two-way analysis of variance (ANOVA) followed by the Bonferroni post hoc test. The remaining parameters were analyzed by using one-way ANOVA followed by Tukey’s post hoc test with a significance level of P < 0.05.

Results

Effect of hibiscetin on acute toxicity

Acute toxicity, oral administration at the recommended safe dose did not exhibit any mortality nor signs of toxicity during the observational period of 14 days. No mortality and other abnormalities such as behavioral, vision, weight, skin, and fur changes were observed. Based on the acute toxicity study, 10 mg/kg of hibiscetin was used in the experiment.

Effect of hibiscetin on Y-maze test

As shown in Figs. 1A and 1B, it was found that the LPS control group significantly (35 ± 1.35; P < 0.0001) reduced SAP and total arm entries compared (6.00 ± 0.53; P < 0.0001) to the control (64 ± 5.23; 13.00 ± 0.53). The data was examined by one-way ANOVA followed by Tukey’s post hoc test. Hibiscetin significantly increased (57 ± 4.95) the SAP (F (3, 20) = 6.714; (P = 0.0026)) and total arm entries (11.00 ± 0.61) (F (3, 20) = 29.72, (P < 0.0001)) as compared to the LPS control group. The per se group did not exhibit any significant effect on SAP (63.00 ± 5.54) and total arm entries (13.00 ± 0.712) as compared to the control group. These findings suggest that hibiscetin improved working memory in LPS-induced rats.

Figure 1 Effect of hibiscetin (A) SAP (B) total arm entries.

One-way ANOVA followed by Tukey’s post hoc test; #P < 0.0001 vs. Control, *P < 0.05, ***P < 0.0001 vs. LPS control (total number of animals n = 24, number of animals in each group = 6).

Effect of hibiscetin on Morris water maze test

As a result of the learning trials, all groups of trained rats were found to be deficient in their escape latency. Training trials showed a significant difference between the LPS- control group and the control group on days 2, 3, 4, and 5, consecutively (P < 0.0001). Two-way ANOVA followed by Bonferroni post hoc test revealed that hibiscetin treatment shows significantly reduced latency-time (F (4, 125) = 215.9, (P < 0.0001)) compared to the LPS-control group on days 2, 3, and 4 (Fig. 2A). The per se group has not shown any significant effect as compared to the control group.

Figure 2 Effects of hibiscetin on (A) escape latency, (B) time spent in quadrant.

One-way ANOVA followed by Tukey’s post hoc test; #P < 0.0001 vs. Control, *P < 0.05, **P < 0.001, ***P < 0.0001 vs. LPS control (total number of animals n = 24, number of animals in each group = 6).

From Fig. 2B, the LPS-control group exhibited a substantial reduction (21.00 ± 2.11; P < 0.001) in time spent in the target quadrant compared to the control group (47.00 ± 3.00). Moreover, one-way ANOVA followed by Tukey’s post hoc test demonstrated that hibiscetin-treated rats exerted a substantial rise (34.00 ± 2.13) in time spent in the target quadrant when compared to the LPS-control group (F (3, 20) = 19.28, (P < 0.0001)). The per se group was no substantial effect (46 ± 3.57) compared to the control group. These findings suggest that hibiscetin improved spatial memory and learning in LPS-induced rats.

Effect of hibiscetin on AChE and ChAT

Figure 3A and 3B shows that the LPS-control group significantly increased (101.2 ± 6.62) AChE activity in their brains (P < 0.0001) compared to the control group (61.0 ± 4.53). One-way ANOVA followed by Tukey’s post hoc test showed that treatment of hibiscetin was substantially reduced (77.37 ± 5.29) AChE activity compared to the LPS-control group (F (3, 20) = 11.27, (P = 0.0002)). These findings suggested that hibiscetin inhibited the content of AChE activity in LPS-induced rats. LPS control group, ChAT activities significantly reduced (62.00 ± 4.95; P < 0.0001) compared to the control group (111.0 ± 6.07). Comparison to the LPS-control group, ChAT levels were increased (86.33 ± 6.68) in hibiscetin treatment (F (3, 20) = 15.18, (P < 0.0001)), indicating that hibiscetin has the ability to restored In the ChAT activity in In LPS-induced rats. Per-se group did not affect AChE (60.00 ± 6.25) and ChAT (110.0 ± 5.98) activity compared to the control group.

Figure 3 Effects of hibiscetin on (A) AChE, (B) ChAT level.

One-way ANOVA followed by Tukey’s post hoc test; #P < 0.0001 vs. Control. *P < 0.05 vs LPS control (total number of animals n = 24, number of animals in each group = 6).

Effect of hibiscetin on oxidative stress and antioxidant markers

A considerable amount of MDA was increased (5.10 ± 0.27) in the LPS-control group (P < 0.0001) compared to the control group (2.30 ± 0.27). One-way ANOVA followed by Tukey’s post hoc test revealed that hibiscetin substantially reduced (3.9 ± 0.27) in MDA levels as associated with the LPS-control group (F (3, 20) = 25.38, (P < 0.0001)). Furthermore, the hibiscetin per se group showed no significant effect (2.20 ± 0.27) compared to the control grou (Fig. 4A).

Figure 4 Outcomes of hibiscetin on (A) MDA, (B) SOD, (C) GSH, (D) CAT level.

One-way ANOVA followed by Tukey’s post hoc test; #P < 0.0001 vs. Control, *P < 0.05 vs. LPS control (total number of animals n = 24, number of animals in each group = 6).

Based on Figs. 4B–4D, LPS-control group had a notable reduction in SOD (18.17 ± 2.55), GSH (35 ± 4.83), and CAT (28 ± 4.83) levels (P < 0.001) in association to control group (SOD 35 ± 2.27 0; GSH 61 ± 3.15; CAT 70 ± 3.15). One-way ANOVA followed by Tukey’s post hoc test demonstrated that hibiscetin significantly increased SOD (29 ± 2.15) (F (3, 20) = 10.75, (P = 0.0002)), GSH (54 ± 3.94) (F (3, 20) = 8.153, (P = 0.0010)), CAT (46 ± 3.80) (F (3, 20) = 22.91, (P < 0.0001)) levels as compared to the LPS-control group. The hibiscetin per se group was no significant effect on SOD (34 ± 2.40), GSH (60 ± 4.75), and CAT (69 ± 4.75) levels compared to the control group. However, hibiscetin improved antioxidant levels and maintained oxidative stress in LPS-induced rats.

Impact of hibiscetin on NO

The LPS-control group increased (33 ± 1.60) NO levels substantially (P < 0.0001) compared to the control group (17 ± 0.69). While compared to the LPS-control group, substantially reduced (27 ± 1.5) NO levels in hibiscetin treatment by using one-way ANOVA followed by Tukey’s post hoc test as (F (3, 20) = 43.37, (P < 0.0001)). There was no significant effect (16.50 ± 0.69) of hibiscetin per se group (Fig. 5).

Figure 5 Outcomes of hibiscetin on NO level.

One-way ANOVA followed by Tukey’s post hoc test; #P < 0.0001 vs. Control, *P < 0.05 vs. LPS control (total number of animals n = 24, number of animals in each group = 6).

Effect of hibiscetin on neuroinflammatory markers

The concentrations of IL-1β (68 ± 4.83), IL-6 (90 ± 4.83), and TNF-α (155 ± 6.35) were substantially increased (P < 0.001) in LPS-control group in contrast to the control group IL-1β (35 ± 3.13), IL-6 (50 ± 3.15), TNF-α (85 ± 5.10). Treatment with hibiscetin lowered, these enhanced amounts to a substantial extent of IL-1β (51 ± 3.94) (F (3, 20) = 6.841, (P = 0.0024)), IL-6 (65 ± 3.94) (F (3, 20) = 19.44, (P < 0.0001)), and TNF-α (127 ± 6.15) (F (3, 20) = 35.66, (P < 0.0001)) compared to LPS-control group (Figs. 6A–6C). Hibiscetin per se is not associated with any effect on IL-1β (34 ± 3.13), IL-6 (51 ± 4.75), and TNF-α (84 ± 5.45) compared to the control group. Overall, hibiscetin restored the neuroinflammatory levels in LPS-induced rats.

Figure 6 Outcomes of hibiscetin on (A) IL-1β, (B) IL-6 (C) TNF-α (D) NF-κB level.

One-way ANOVA followed by Tukey’s post hoc test; #P < 0.0001 vs. Control, *P < 0.05 vs. LPS control (total number of animals n = 24, number of animals in each group = 6).

Effect of hibiscetin on NF- κB

Referring to Fig. 6D it was observed that the LPS-control group activated the NF-κB (15.80 ± 1.60) considerably (P < 0.0001) in comparison with the control group (5.8 ± 0.69). However, hibiscetin treatment substantially reduced (10.60 ± 1.36) in NF-κB activity compared to the LPS- control group (F (3, 20) = 17.44, (P < 0.0001)). No significant effect was observed from the hibiscetin per se group (5.5 ± 0.69). These findings suggest that hibiscetin inhibited NFκB activity.

Effect of hibiscetin on BDNF

We assessed the expression levels of BDNF in rat’s brain. Compared to the control group (19.46 ± 0.74), the LPS-control group exhibited significantly (11.33 ± 0.92) declined BDNF expressions (P < 0.0001).

One-way ANOVA followed by Tukey’s post hoc test revealed that expression levels of BDNF were significantly restored in hibiscetin administration (16.07 ± 0.99) compared to the LPS-control group (F (3, 20) = 39.65, (P < 0.0001)) (Fig. 7A). No substantial effect was seen in the hibiscetin per se group (20.50 ± 0.60) compared to the control group. These findings indicate that hibiscetin increased BDNF expression levels which is important for the formation and consolidation of long-term memories.

Figure 7 Outcomes of hibiscetin on (A) BNDF and (B) Caspase-3 level.

Mean ± S.E.M. (n = 6). One-way ANOVA followed by Tukey’s post hoc test; #P < 0.0001 vs. Control, **P < 0.001 vs. LPS control (total number of animals n = 24, number of animals in each group = 6).

Effect of hibiscetin on caspase-3

The LPS control group significantly increased (5.90 ± 0.19) caspase-3 levels in rats (P < 0.0001) compared with the control group (3.00 ± 0.18) (Fig. 7B). One-way ANOVA followed by Tukey’s post hoc test that hibiscetin administration led to significant downregulation (4.90 ± 0.19) in caspase-3 levels compared to LPS-control group (F (3, 20) = 60.45, (P < 0.0001)). Nevertheless, the hibiscetin per se group does not appear any substantial effect (2.90 ± 0.18) compared to the control group. These results indicate that hibiscetin inhibits caspase-3 activity.

Discussion

Memory impairment and behavioral abnormalities are the cardinal manifestations of AD, PD, HD, and many other neurological disorders. The present study demonstrated that LPS can induce memory impairment in rats, accompanied by changes in behavior, learning and memory, biochemical parameters, neuroinflammatory markers, oxidative stress, and neurotransmitter levels. Locomotor functions and behavioral changes are primary functions of the brain and its neurotransmitters, and disruption of these parameters leads to memory impairment (Amraie, Pouraboli & Rajaei, 2020; Alshehri & Imam, 2021; Ammari et al., 2018; Amooheydari et al., 2022).

The brains of rats with LPS-induced neuroinflammation exhibited elevated levels of AChE, oxidative stress markers, inflammatory markers, and BDNF, which contributed to the observed behavioral abnormalities. Treatment with hibiscetin improved both behavioral and biochemical parameters (Azmand & Rajaei, 2021; Babaei et al., 2022; Arab et al., 2022; Beheshti et al., 2019). Specifically, hibiscetin reduced inflammatory markers and BDNF concentrations in rat brains, while also restoring endogenous antioxidant status (Careena et al., 2018; Jamali-Raeufy et al., 2021; Kang et al., 2022; Kesika et al., 2021).

The behavioral paradigm of rats was assessed in the present study using the Y-maze and Morris water maze tests. Prior to LPS and hibiscetin treatment, the rats were thoroughly trained in both apparatuses. In the Y-maze test, LPS-injected rats exhibited a decrease in SAP and total arm entries. Notably, hibiscetin treatment improved SAP and total arm entries, as evidenced by behavioral evaluations in rodents (Keymoradzadeh et al., 2020; Khajevand-Khazaei et al., 2018; Khan et al., 2019).

Following LPS injection, a significant increase in escape latency in rats was observed, indicating a decline in cognitive function. However, hibiscetin treatment subsequently restored escape latency, suggesting a restoration of cognitive function. Additionally, due to impaired cognitive function, LPS-injected rats spent less time in a specific quadrant of the Morris water maze test. The time spent in the target quadrant, which reflects improved cognitive function, increased after hibiscetin treatment (Sadraie et al., 2019; Rocha-Gomes et al., 2022).

Acetylcholine (ACh) plays a vital role in memory enhancement, and reduced levels of ACh lead to memory loss. ChAT synthesizes ACh, while AChE degrades it. LPS decreased ChAT levels and increased AChE levels, which indirectly lowered ACh levels and resulted in memory deficits. Hibiscetin reduced AChE levels and elevated ChAT levels, maintaining normal ACh concentrations in the brain. This helped to improve the recovery of impaired memory by counteracting the elevated AChE levels and raising the declining ChAT levels. (Alshehri & Imam, 2021; Ammari et al., 2018; Amooheydari et al., 2022; Ma et al., 2022; Mao et al., 2021; Marefati et al., 2020, 2021; Mihaylova et al., 2021; Momeni, Ebadi & Sadegh, 2021; Moosavi Sohroforouzani et al., 2020).

Elevated oxidative stress and the production of free radicals are thought to contribute to memory loss. MDA is considered a marker of oxidative stress. MDA levels in LPS-injected rats increased significantly, but hibiscetin substantially reduced these levels. SOD, GSH, and CAT are antioxidant enzymes that prevent the formation of free radicals and oxidative stress-induced damage (Kshirsagar et al., 2021). A study found that LPS-treated rats had decreased levels of SOD, GSH, and CAT antioxidant enzymes. In the present study, hibiscetin restored these levels almost to normal (Kshirsagar et al., 2021; Shu et al., 2020; Singh & Bansal, 2021; Sohroforouzani et al., 2022; Syeda et al., 2021).

Following LPS injections, neuroinflammatory cytokine levels increased significantly, leading to inflammation of brain cells and impaired memory. Hibiscetin reduced neuroinflammatory marker concentrations. NF-κB, a transcription factor that regulates the expression of inflammatory genes, also played a role in producing the inflammatory response. LPS-injected rats exhibited increased NF-κB levels. In contrast, hibiscetin significantly decreased these levels (Keymoradzadeh et al., 2020; Kshirsagar et al., 2021; Beheshti et al., 2019; Bishnoi, Kavaliers & Ossenkopp, 2023).

LPS administration significantly elevates nitric oxide (NO) levels in the brain, which is directly correlated with learning and memory. Hibiscetin treatment reduces brain NO levels and improves learning-associated memory (Keymoradzadeh et al., 2020; Khajevand-Khazaei et al., 2018; Khan et al., 2019; Zhang et al., 2018). BDNF plays a vital role in memory enhancement by increasing the brain’s capacity for information storage. Hibiscetin treatment restored normal BDNF levels in rats that had been decreased by LPS treatment (Syeda et al., 2021; Zhang et al., 2018; Tang et al., 2019; Wang et al., 2018; Xie et al., 2021; Zhang et al., 2021). Caspase-3 is a proteolytic enzyme that can trigger apoptosis, or programmed cell death, in brain cells. When rats were administered LPS, a bacterial endotoxin, their caspase-3 activity was significantly upregulated. Hibiscetin, a flavonoid compound, was able to prevent the death of brain cells by downregulating elevated caspase-3 activity. Overall, hibiscetin itself did not appear to have any significant effects on rodents (Sadraie et al., 2019; Schilling et al., 2021; Norouzi et al., 2019; Pierre et al., 2022; Pusceddu et al., 2022; Rocha-Gomes et al., 2022; Shu et al., 2020; Singh & Bansal, 2021; Sohroforouzani et al., 2022; Rahim et al., 2021).

Hibiscetin inhibited the proinflammatory cytokine pathway, oxidative stress, BDNF, and caspase-3 activity, while improving endogenous antioxidant levels (CAT, GSH, and SOD). Hibiscetin also reduced elevated AChE levels and increased ChAT concentration, resulting in increased ACh levels. These underlying molecular mechanisms likely underlie hibiscetin’s potent antioxidant effects by modulating the aforementioned pathways in the LPS-injected memory deficit paradigm. Further studies, including immunohistochemistry, molecular studies (e.g., western blotting and RT-PCR), tissue immunohistochemistry, pharmacokinetic and pharmacodynamic studies, and other genetic models, should be conducted to confirm this mechanism. The study’s limitations include its short duration and the use of a limited number of animals.

Conclusion

The present study demonstrated the efficacy of hibiscetin in attenuating memory impairment induced by LPS in rats. The study established that hibiscetin was able to restore behavioral and cognitive function in LPS-injected rats. Additionally, hibiscetin was effective in controlling various parameters associated with memory impairment, such as AChE, ChAT activity, antioxidant enzyme levels, and neuroinflammatory parameters. These findings warrant further investigation in humans to validate and expand upon the results. Given its natural origin, hibiscetin may prove to be a safe and effective therapeutic intervention for memory impairment, offering a promising alternative to synthetic drugs. This natural compound also has potential as a valuable ingredient in the herbal industry for the development of suitable formulations.

Supplemental Information

Supplemental Information 1 The hypothetical mechanism of hibiscetin attenuates lipopolysaccharide-evoked memory impairment.

Click here for additional data file.

Supplemental Information 2 Raw data.

Click here for additional data file.

Supplemental Information 3 ARRIVE 2.0 Checklist.

Click here for additional data file.

Additional Information and Declarations

Competing Interests

Author Contributions

Animal Ethics

Data Availability

The authors declare that they have no competing interests.

Sadaf Jamal Gilani analyzed the data, authored or reviewed drafts of the article, and approved the final draft.

May Nasser Bin Jumah analyzed the data, authored or reviewed drafts of the article, and approved the final draft.

Farhat Fatima performed the experiments, analyzed the data, prepared figures and/or tables, authored or reviewed drafts of the article, and approved the final draft.

Fahad A. Al-Abbasi analyzed the data, authored or reviewed drafts of the article, and approved the final draft.

Muhammad Afzal analyzed the data, authored or reviewed drafts of the article, and approved the final draft.

Sami I Alzarea analyzed the data, authored or reviewed drafts of the article, and approved the final draft.

Nadeem Sayyed performed the experiments, analyzed the data, prepared figures and/or tables, authored or reviewed drafts of the article, and approved the final draft.

Muhammad Shahid Nadeem analyzed the data, authored or reviewed drafts of the article, and approved the final draft.

Imran Kazmi conceived and designed the experiments, authored or reviewed drafts of the article, and approved the final draft.

The following information was supplied relating to ethical approvals (i.e., approving body and any reference numbers):

Institutional Animal Care and Use Committee (IACUC/TRS/PT/022/027) in India

The following information was supplied regarding data availability:

The raw data is available in the Supplemental Files.

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
