# Peer review of "Hibiscetin attenuates lipopolysaccharide-evoked memory impairment by inhibiting BDNF/caspase-3/NF-κB pathway in rodents"

_PeerJ, doi:10.7717/peerj.16795_

## Round 0.1 · original submission · Major Revisions

I have now received the reviewers' comments on your manuscript. They have suggested some major revisions to your manuscript. Therefore, I invite you to respond to the reviewers' comments and revise your manuscript.

Reviewer 1 ·

Basic reporting

The authors of this study investigated the neuroprotective potential of hibiscetin against lipopolysaccharide (LPS)-induced memory imapirment in rodents. They found that hibiscetin was effective in improving memory function in LPS-treated animals, and that this effect was associated with reductions in oxidative stress and neuroinflammation. authors used animal model of LPS-induced memory deficits. They used a well-established behavioral test to assess memory function and used an array of biochemical and neuroinflammatory markers to assess the effects of hibiscetin. However, the novelty of the findings is limited, as the antioxidant and anti-inflammatory properties of flavonoids are well-known. I would recommend that the article be accepted for publication, but after the authors make the suggested changes as listed below.

Experimental design

1. The specific source of LPS should mentioned, the species from which it was extracted ( Catlogue number, lot number etc.) as the composition may change with species to species and properties may vary with different batches of extraction. This will ensure the reproducibility of the results.
2. Source of hibiscetin also must be added. Was it commercially available, or was it extracted and purified? If it was extracted and purified, please add the details of the process.
3. Add more references to the introduction section, where stating the goal, and while discussing the results. Few of the references are mentioned below (mostly from the authors themselves, suggesting neuroprotective, anti-inflammatory and antioxidant properties of the hibiscetin)
a. https://doi.org/10.1016/j.biopha.2020.110303
b. https://doi.org/10.3390/molecules28031402
c. https://core.ac.uk/download/pdf/287236497.pdf
d. https://doi.org/10.1016/j.jsps.2022.09.016
4. Animal group numbers (line 127) states that n=30, but data is for 24 animals (6x4). Kindly correct them or explain the discrepancy
5. Correct the unit mentioned in line number 170. (Detection of Acetyl-cholinesterase (AChE) and choline-acetyltransferase (ChAT)

Validity of the findings

All underlying data is provided.

Additional comments

Please improve the language, typos, and punctuation in the following lines: 79-80, 83-84, 127, 185, and 222.

·

Basic reporting

1. The English language should be improved to make is more comprehendible. The text (including abstract, introduction and discussion) is full of grammatical errors. Some examples include lines 77,78,79.

2. The novelty of the manuscript is not explained at all.

3. The study lacks the overall big picture. It is not clear what hypothesis is being tested and the rationale behind doing the experiments. The introduction needs to be rewritten with a clear emphasis on the big question addressed in the manuscript, the overall hypothesis tested and how this study answers the unanswered question.

4. The results are stated like figure legends and lacks hypothesis and rationale. The result section should be expanded and the relevance of the results obtained should be clearly stated.

5. The discussion focuses on the effects of LPS rather than focusing on Hisbiscetin which appears to be the primary focus of the study. Therefore, the discussion should be edited heavily.

Experimental design

1. The methods and figure legends should clearly state how many animals were used for each experiment and how many times each experiment was repeated.

2. What was the age of the rats when LPS/Hisbiscetin was administered?

3. The authors should include Pharmacokinetics/Pharmacodynamics experiments with Hisbiscetin. Does Hisbiscetin cross the blood spinal cord barrier? The data supporting this should be included.

4. The authors should also include data to show the effect of Hisbiscetin on the survival of the rats. This will add depth to the results and improve the quality of the manuscript.

5. Replot the bar graphs to show the individual data points in the graph.

7. Please expand the abbreviations used in the manuscript (YMT, MWMT etc.)

Validity of the findings

The novelty of this manuscript is not clear. The authors need to focus on explaining what is it that makes their work novel.

---

## Round 0.2 · Major Revisions

Thank you for the revision. However, there are still concerns that prevent me from accepting the revised paper. Please pay attention to the reviewers' comments and respond to them carefully.

·

Basic reporting

The authors have responded to some of my previous comments, however, there are several major issues in the revised manuscript that still needs to be addressed:

1. The authors state "While the disease can occur at any age, it is most often diagnosed in elderly people as the disease progresses. A common parameter of such conditions is memory impairment, and patients are even unable to perform their routine tasks as a result of the disease." Which disease/s are the authors referring to, just AD or AD, HD, PD?

2. Some sentences are confusing. For example

"Moreover, hibiscetin previously reported antioxidant activity (27), against streptozotocin-induced diabetes (28), favourable effect on rotenone-induced Parkinson's disease and 3-nitro propionic acid-induced Huntington’s Disease (29, 30) via inhibiting inflammatory and oxidative stress."

"If previous research has not extensively explored the first study to report the effects of hibiscetin in this specific model, the study's findings introduced new insights into its potential neuroprotective properties."

"The chemical produced a condition causing a loss in memory occurred (44)".

3. instead of presenting the results as "Acute Toxicity", "Y maze test" , "NF-kB" etc, the authors should expand the subheadings and make it more descriptive like " The effect of hibiscetin on NF-kB levels" etc

4. The English language in the manuscript needs to be majorly improved.

5. The authors have not addressed my previous comment stating that - The results are stated like figure legends and lacks hypothesis and rationale. The result section should be expanded and the relevance of the results obtained should be clearly stated.

Experimental design

N/A

Validity of the findings

N/A

---

## Round 0.3 · accepted · Accept

Many thanks for addressing all the issues.

·

Basic reporting

The authors have addressed all my comments.

Experimental design

N/A

Validity of the findings

N/A